# Positive Effect of Ecological Restoration with Fabaceous Species on Microbial Activities of Former Guyanese Mining Sites

**DOI:** 10.3390/molecules27061768

**Published:** 2022-03-08

**Authors:** Ewan Couic, Alicia Tribondeau, Vanessa Alphonse, Alexandre Livet, Noureddine Bousserrhine

**Affiliations:** 1Water, Environment and Urban Systems Laboratory (Leesu), Department of Biogeochemistry, University of Paris-Creteil, 94010 Creteil, France; vanessa.alphonse@u-pec.fr (V.A.); livet@u-pec.fr (A.L.); bousserrhine@u-pec.fr (N.B.); 2National Museum of Natural History, 75005 Paris, France; alicia.tribondeau@edu.mnhn.fr

**Keywords:** gold mining, biogeochemistry, ecological restoration, fabaceous species, microbial community, catabolic diversity

## Abstract

Understanding ecological trajectories after mine site rehabilitation is essential to develop relevant protocols adapted for gold mining sites. This study describes the influence of a range of mine site rehabilitation and revegetation protocols on soil physicochemical parameters and microbial activities related to carbon, nitrogen and phosphorus cycles. We sampled soil from six rehabilitated mining sites in French Guiana with different plant cover (herbaceous, Cyperaceous, monoculture of *Clitoria racemosa* and *Acacia mangium* and association of *C. racemosa* and *A. mangium*). We measured the mineralization potential of organic matter by estimating the mineralization of carbon, nitrogen and phosphorus and the microbial catabolic diversity balance. The results showed an improvement in the quality of organic matter on revegetated sites with tree cover. On restored sites with fabaceous species, the microbial biomass is three times higher than non-restored sites, improving the rates of organic matter mineralization and restoring the catabolic diversity to the level of natural Guyanese soils. These results confirm that the establishment of fabaceous species under controlled conditions significantly improves the restoration of microbial communities in mining soils.

## 1. Introduction

Gold mining has been a key activity in Guyana’s economy for nearly 150 years [1] and led to significant soil degradation, and soil microbial diversity and activity tended to decrease [2]. This alteration of microbial communities leads to a disruption of biogeochemical cycles, which are at the root of ecosystem functioning [3]. Before 2006, alluvial gold panning mainly used mercury because of its amalgamating properties, thus improving extraction efficiency. Decades of mining and mercury use have had serious consequences on Guyana’s tropical forest, such as deforestation, pollution of water systems, and mercury contamination of local populations [1]. To limit the damage caused by mining, operators are now obliged to rehabilitate mining sites by terracing the soil to homogenize soil structure and texture. After rehabilitation, gold miners are encouraged, but not obliged, to carry out ecological restoration of mining sites by reintroducing plant species [4]. Currently, revegetation is considered a management tool for the sustainable development of the mining industry and soil recovery [2]. In the ecology of restoration, fabaceous plants are known to be efficient and effective and the literature reports numerous studies on their use in soil remediation and restoration, particularly in Guyanese mining sites [5]. The use of legumes in ecological restoration protocols is mainly motivated by their ability to develop symbiotic relationships with nitrogen-fixing bacteria and arbuscular mycorrhizae [6,7]. Among legumes, *Acacia mangium* [8] and *Acacia leucana* [9] are often used because of their rapid growth in nutrient-poor environments and their ability to improve soil fertility. However, to date there is little evidence of the effect of ecological restoration with legumes on soil microbial communities, their diversity and activities in relation to biogeochemical cycles.

Ecological restoration of mining sites would also limit erosion and subtoxic conditions favorable to the mobility of toxic elements, such as mercury. In contrast, non-remediated sites are more sensitive to erosion, particle transfer into the water system, and increased mobility of mercury at these sites, which can increase the potential for methylation and bioaccumulation in the food chain [10]. Mastery of rehabilitation and ecological restoration techniques is therefore essential for sustainable development in Guyana.

The effectiveness of ecological restoration can be estimated with physical (bulk density [11], aggregate stability [12]), chemical (soil pH [13], quantity and quality of organic matter [14]) and biological (microbial activities, plant biomass, genetic diversity [15,16]) methods. Among the biological methods, taking into account the telluric microbial activities makes it possible to account for the degradation and mineralization activities of organic matter and the main soil nutrients, and more broadly for biogeochemical cycles. As these activities support soil quality and its functional diversity, the use of microbial activities as bio-indicators of ecological restoration makes it possible to assess the effect of different mining site restoration protocols.

Nevertheless, ecological restoration monitoring projects are difficult to implement in French Guiana and as a result, there is currently little information on the fate of soil and microbial communities during ecological restoration processes in French Guiana. This lack of data makes it difficult to select the different plant, herbaceous or tree cover and limits the choice of fabaceous species for the restoration of ecosystems. 

The aim of this study is to highlight the influence of different types of plant cover, such as (1) spontaneous pioneer vegetation belonging to the genus of *cyperaceae* and *lycopodiella* sp., and (2) a forest cover with different species of the Fabaceae genus, on microbial activities developed after the rehabilitation of mining sites in the Belizon and Yaoni regions. The main hypothesis of this work is that controlled restoration with fabaceous species is more effective than spontaneous vegetation to cover microbial activities of organic matter mineralization, which are at the origin of major biogeochemical cycles. We also assume that monitoring microbial activities during ecological restoration is a relevant indicator for describing the state of the ecosystem. To evaluate the functionality of the biogeochemical cycles on the restored plots, measurements of C, N and P mineralization and functional microbial diversity were carried out. These results could potentially validate the use of legumes in the ecological restoration process based on microbial indicators linked to the mineralization of organic matter and the functional richness of communities. 

## 2. Results

### 2.1. Soil Properties

Granulometry results (Table 1) showed heterogeneous textures between the rehabilitated sites. The Sp2, Cli, Aca and Mf sites were predominantly sandy with values between 36.4 and 71.1% of sand, with significantly different fine sand/coarse sand ratios between the sites. The Sp1 site was silty clay and the Sp3 site was silty. The contents of organic matter, C, N and P, showed contrasting results between the non-restored (NR) and restored ^®^ sites. For macronutrients, the concentrations were between 3.52–22.88, 0.14–1.81, 0.13–0.76 g·kg^−1^ for total carbon, nitrogen, and phosphorus, respectively, with concentrations always higher for the R sites, with a maximum always reached for the Mf site.

Overall, the Aca and Mf sites were significantly richer in dissolved organic carbon, Ctot, and Ntot, but the restored Cli site had organic matter contents similar to those measured at the Sp3 site. The two unrestored sites Sp1 and Sp2 were overall the poorest sites in organic matter and total organic carbon. The C/N ratios showed significant differences with high ratios for the Sp2 and Sp3 sites and lower for the Aca and Mf sites with values 12.6 in average. Regarding iron, aluminum and mercury contents, the concentrations were rather homogeneous between the different restored and unrestored sites and reflected concentrations consistent with tropical soils with respective ranges of 44.56–113.3 g Fe kg^−1^, 16.4–53.88 g Al kg^−1^ and 0.23–0.48 µg Hg kg^−1^. The pH was generally acidic between the different sites, with a more marked acidity for the Sp3, Aca and Mf sites with pH < 5.

### 2.2. C, N, P Mineralization and Microbial Biomass

Table 2 illustrates the parameters related to organic matter mineralization, soil respiration and microbial biomass in the six soils studied. An increase in MBC was observed across the level of restoration (NR and R) and plantation types (Cli, Aca and Mf). The highest MBC was registered for the Mf site (1103 mg kg^−1^), followed by the Aca site (909 mg kg^−1^) and the Cli site (845.2 mg kg^−1^). For the non-restored soils, MBC was under 400 mg kg^−1^.

Soil respiration (Cmin) measurements showed very significant differences between NR and R with values between 4.45 and 46.58 µg C g^−1^day^−1^ soils, with an increase in organic carbon mineralization activity by a factor of 7 to 9 for the R sites, with the highest values measured for the Mf site. This trend was also illustrated for nitrogen with values between 0.14 and 2.7 µg N g^−1^day^−1^ and phosphorus mineralization with values between 0.017 and 0.065 µg P g^−1^day^−1^. For nitrification, the R sites had a very significantly higher mineralizing activity than the NR sites, by a factor of more than 50. Ammonification, although significantly higher for the R sites, showed smaller differences between the R and NR sites. Finally, for phosphorus mineralization, the results also showed a significantly higher mineralizing activity for the R sites. In terms of carbon, nitrogen and phosphorus mineralization, site Mf was consistently the site with the highest microbial activity.

### 2.3. Catabolic Diversity

The respective ratios of each category of consumed substrates to the total number of substrates in the Biolog Ecoplate are shown in Figure 1. The total value of substrates consumed corresponds to the PFR (see Table 3) and the average of substrates consumed per substrate category is shown in the histograms for each of the study sites.

Figure 1 shows that the relative proportions of consumed substrates are homogeneous between sites Sp1 and Sp2, and Aca and Mf, and that each category of substrates was consumed. For the Sp3 and Cli sites, amines were not consumed during the experiment and amino acids were consumed in a higher proportion. Although the proportions of substrates consumed within each category are relatively equivalent, the number of positive wells for the Aca and Mf sites showed very significant results between unrestored and restored sites, illustrating that the microbial communities had a higher metabolic activity for restored sites. The Cli site, which was younger than the Aca and Mf sites, showed results intermediate between the NR sites and the Aca and Mf sites in terms of substrate metabolism.

The biological properties related to catabolic diversity and calculated from the previous results are shown in Table 3. The overall AWCD factor showing substrate metabolism indicated significantly higher metabolic activity for the N sites and very low metabolic activity for the Sp1 and Sp3 sites, with respective values of 0.08 and 0.09, which was coupled at these sites to an average degradation of <4 substrates. For the restored site, Mf, the AWCD reached a maximum value of 0.49 which gives a factor of 5 in microbial growth between the Sp and Mf sites.

Catabolic diversity was significantly higher for the R sites, with higher diversity for the Aca and Mf sites with values above 0.30 for Cli site and 0.62 for the Mf site. Again, the lowest values were determined for sites Sp1 and Sp2 with an average of catabolic diversity of 0.12. The Shannon Index (H) was significantly lower for sites Sp1 and Sp2 and an index greater than 3 was measured for the restored site, Mf. Finally, the regularity index showed results close to 1 for the Cli, Aca and Mf sites and low values for the Sp1 and Sp2 sites. Overall, the restored site, Mf, showed significantly higher metabolic and functional diversity indicators than the other sites, with very significant differences between the NR and R sites.

### 2.4. Effect of Ecological Restoration on Microbial Parameters

Table 4 shows the results of the ANOVA tests performed on all the experimental data. Two factors were retained to carry out the ANOVA. The first factor is the site effect, where significant differences between the six sites studied were tested. The second is the effect of the level of restoration, which allows us to test for significant differences between the two groups of restored (R) and non-restored (NR) sites. All statistical tests show significant differences when considering the factors sites or level of ecological restoration. 

### 2.5. Overall Variability of the Database

Table 5 highlights the results of the correlation tests for all microbial parameters in the six soils studied. The best indicator of the diversity and functional richness of the microbial communities seems to be the microbial biomass carbon (MBC) with significant r^2^ of 0.89, 0.85 and 0.84 for the PFR, Shannon Index and AWCD, respectively.

Among the indicators of organic matter mineralization activities, carbon mineralization is significantly correlated with the indicators of diversity and functional richness with significant r² of 0.82, 0.82 and 0.74 for the PFR, Shannon Index and AWCD, respectively. Phosphorus mineralization is also significantly correlated with r² of 0.93, 0.89 and 0.85 for the PFR, Shannon Index and AWCD, respectively.

Overall nitrogen mineralization is not significantly correlated with diversity indicators, but ammonification is significantly correlated with r^2^ of 0.77, 0.77 and 0.72 for the PFR, Shannon Index and AWCD, respectively.

Except for nitrification, organic matter mineralization rates are significant indicators of diversity and functional richness.

The principal component analysis presented in Figure 2 explains a significant proportion of the experimental variability in the data from this study. Axis 1, which explains 52% of the variability, is mainly driven by diversity and functional richness indices, nutrient concentrations, and organic matter mineralization rates as well as the percentage of plant cover. Mercury concentrations are also among the most explanatory variables in axis 1. The criterion of cessation of mining activity on axis 1, illustrating the age of the sites, is not a very explanatory criterion for the variability. Axis 2, which explains 20% of the variability, is mainly driven by soil texture and Al and Fe concentrations and is not correlated with the biological variables. 

The figure on the right allows the samples to be grouped according to the different sites. The PCA shows very significant differences between the sites (*p* < 0.001). the distance between the clusters being proportional to the differences between the samples, the figure shows that the sites Mf and Aca are globally very close and that the third restored site Cli is globally close to the unrestored site Sp2. The last two sites, Sp1 and Sp3, which showed quite similar results on the overall results, are also very close on the PCA and behave in a similar way.

## 3. Discussion

### 3.1. Positive Effect of Ecological Restoration on Nutrient Content in Soil

Firstly, it is important to remember that the mining activity on these sites initially destroyed the surface horizons after mechanical leaching of the soil particles. The reconstitution of the soils is carried out after the end of the exploitation and the earthwork is carried out in a heterogeneous way, which could explain the important differences in terms of texture of the old mining soils. In this context it might be more appropriate to refer to these soils as anthroposols, or soils reconstituted by human activity, rather than to compare them directly to tropical ferralitic soils. It is nevertheless interesting to note that the concentrations of iron and aluminum oxides, indicative of the formation of tropical soils, were consistent with the natural concentrations in Guyanese soils [17,18,19]. In other words, mining activity seemed to have altered only the amount of organic matter, as evidenced by the very low macronutrient concentrations of the unrestored sites, and the soil texture. 

Restored soils had homogeneous texture, while the texture of NR soils was mainly varied. The differences in textures indicated that rehabilitation was performed differently, and this finding was supported by Ekamawanti, who showed high soil heterogeneity in rehabilitated sites [20]. These different rehabilitation strategies might have an impact on microbial activities. Hou reported that there is an interaction between soil properties and the structure and function of soil microbial communities, particularly in the decomposition of organic matter [2]. The relatively homogeneous structure of the R sites (Table 1) ensured an adequate water balance for vegetation cover, and the very coarse texture of the Sp2 site and the very fine textures of the Sp1 and Sp2 sites limit the recovery of vegetation cover.

Regarding mercury levels in the studied soils, the results showed an interesting trend as highlighted in the PCA (Figure 2). The higher concentration of Hg for R sites, with 0.39 µg kg^−1^, than NR sites, with 0.24 µg kg^−1^, could suggest that Hg is trapped in the soil, mainly due to the presence of plantations. Several studies have reported that plantations are an essential means to stabilize a bare area and to minimize the pollution from mine sites [21,22], in particular, by limiting the processes of erosion and the transfer of mercury particles to the water system and by drying the soil, which limits anaerobic conditions favorable to mercury methylation. However, mercury measurements at the sites are very heterogeneous, and additional measurements would be needed to confirm this claim.

The contents of C, N, P and TOC were significantly higher in the R sites than in the NR sites. Many studies have shown that soil nutrient content is closely correlated with changes in the vegetation cover [23,24,25], and vegetation age affects the vertical distribution of microorganisms [2]. These results also confirm the effectiveness of a cover of fabaceous species in recovering soil organic matter stocks at rehabilitated sites in Guyana. A major point of this study can also be illustrated with the C/N ratios between NR (Sp2 and Sp3) and R soils. The presence of fabaceous species and their ability to fix atmospheric nitrogen could also improve the quality of organic matter, as shown by the values of C/N < 15 at the Aca and Mf sites compared to C/N > 20 for the Sp2 and Sp3 sites.

Revegetation with fabaceous plants is a widely used technique to restore soil fertility by covering the soil with vegetation and permitting entry of nitrogen in the biological cycle [25,26]. Several studies have shown that the type of vegetation cover has an influence on improving the physicochemical quality of mined soils, as well as on the overall level of ecological restoration. 

In our study, vegetation played a significant role in improving physicochemical properties, especially in the mixed fabaceous (Mf) site. Despite an equivalent age for the restored Aca and Mf parts, the results clearly illustrated that the sites restored with an association of Fabaceae at the start of the ecological restoration process developed higher quality indicators (organic matter) than for the monocultures. Due to this recovery of soil organic matter, the most likely hypothesis would be that the Mf sites show higher biogeochemical indicators of microbial activity than the other restored sites described by Schimann et al. [27].

### 3.2. Positive Effect of Ecological Restoration on Soil Respiration and Organic Matter Mineralization

To assess the soil quality in rehabilitated and regenerated mining sites, microbial carbon biomass and mineralization of macronutrients are relevant tools [27,28]. According to Ross et al. [29], the microbial biomass is a satisfactory estimate of the restoration of the soil microbial population. In our study, the MBC was different for each site, indicating significant effects of the rehabilitation process, and was related to vegetation cover (Table 4) with the greatest effect for the Mf sites. These microbial biomass values remain lower than those recorded in a previous study on natural soils in Guyana and show that the full recovery of microbial density is not yet complete for the restored sites [28].

The differences in MBC between restored soils (Cli, Aca and Mf) are related particularly to the age of the plantations and the species [24,25,30], especially their litter inputs and root exudates [23]. Aca and Mf were planted in 1998, while Cli was planted in 2013. This demonstrates that a continuous increase in MBC with age indicates a continuous recovery of mining soils [26], and that estimated MBC recovery does not need a long period. MBC was higher in R than NR soils. This difference is due to plantations. Several studies reported the positive effect of plantations on microbial biomass [30] through soil organic matter restoration, essential nutrient pools and soil structure. Organic matter stabilizes aggregate structure, increases soil retention capacity and improves nutrient bioavailability [31,32], thus stimulating soil microbial flora [33,34,35]. In addition, it highlighted the role of vegetation in comparison with the non-restored sites with few pools of carbon, nitrogen and phosphorus [24,30,36].

For these nutrients, the nutrient quality of the restored R soils was higher than that of the non-restored sites. These results appear to be directly related to the level of vegetation cover (Table 4). Previous investigations on soil reclamation reported increased nutrient pools following the addition of compost, lime, and vegetation [2].

Concerning microbial activities linked to the functioning of the main biogeochemical cycles, the differences between sites and between ecological restoration methods are very significant (Table 4) and may confirm the role of restoration in the recovery of soil functionalities, as observed in a previous study in experimental restoration sites in Guyanese mining sites [28].

Large differences in the carbon mineralization rate showed the beneficial effect of ecological restoration. R soils were characterized by a higher rate of C, N, and Pmin in comparison with NR soils. These differences between the two restoration treatments also highlighted the major effect of the restoration process achieved by combining different plant species as shown by the higher mineralization rates for the Mf sites. This improvement in microbial functionality could be attributed to differences in total microbial biomass between mine sites. These results also confirmed the positive effect of plant cover in stimulating microbial activities related to the turnover of organic matter [2,28].

Revegetation induced increases in the total N-mineralization rates by 2.03, 1.55 and 2.7, respectively, for Mf, Cli and Aca, in comparison with lower values in non-restored soils. However, total Nmin was higher in Cli sites than Aca, which might be because of slow litter decomposition and lower N return through litterfall by *Acacia* spp. compared to *C. racemosa*, as reported by Singh et al. [25], who found the same difference between *A. procera* and *A. lebbeck*. These results could suggest that the different Fabaceae species and their associations are not all equally effective in restoring the functionalities of the nitrogen cycle in rehabilitated soils. It is possible that different covers of fabaceous species may induce variations in the diversity of microbial communities in the nitrogen cycle [37].

As highlighted by Table 5, the restoration factor, and thus establishment of a vegetation cover, was an explanatory factor for each of the microbial parameters determined in this study. These differences between restored and non-restored sites highlighted that active restoration process is an important step for effective restoration and that the Fabaceae *Clitoria racemosa* and *Acacia mangium* are good candidates for effective restoration of biogeochemical cycles in mining soils.

### 3.3. Positive Effect of Ecological Restoration on Catabolic Diversity and Diversity Index of Soil Microbial Communities

Vegetation cover has an influence on the activity of microbial communities and their ability to provide compounds for biogeochemical cycles. However, several studies have demonstrated that the microbial communities on anthropized sites, such as old mine rehabilitated soils, can be altered. AWCD, catabolic diversity, PFR, Shannon and EH measurements indicated that R sites were richer than NR sites in terms of diversity and biological activity, and these differences might be attributed primarily to vegetation [2,27,36] and secondarily to texture and soil properties [27].

The AWCD value in an EcoPlate™ well is an important index of microbial functional diversity because it represents the capacity of soil microorganisms using different carbon sources [38]. The increase in AWCD of R soils indicated a higher rate of carbon source use and higher functional diversity than NR soils. This might be attributed to plantations, which stimulate the microorganisms. A positive correlation between AWCD, MBC and total C, N, and P suggests that available C, N, and P are key factors influencing soil microbial functional diversity, which is improved by the nature of the vegetation cover [38]. Li et al. [23] showed the effect of vegetation as a significant factor for the development of microbial soil properties in many reclaimed treatment studies. In addition, it has already been shown that a rich vegetation cover, including fabaceous species, could stimulate microbial activities [23], which is in agreement with our results. Catabolic diversity and PFR parameters were significantly different between the levels of restoration. The same pattern was observed for Shannon and EH, indicating that the greatest catabolic diversity existed under R soils and that the least diversity existed under NR soils. These results support the hypothesis that the establishment of a plant cover favors microbial activities and functional diversities by improving the physicochemical properties of the soil [39]. It is important to note that the catabolic diversity measured at the Sp2 and Sp3 sites was very similar to that obtained by Harris (2009) [17] from natural Guiana soils. These results could, therefore, illustrate a return to a balance in catabolic diversity and some biogeochemical functionality.

To determine whether the level of ecological restoration had an impact on the structure of microbial communities, we conducted a PCA (Figure 2) to estimate the variance of the dataset explained and to determine the global difference between sites and restoration level. This PCA was significant (*p* = 0.001) and explained a high percentage of total variance, while the eigenvalues for axes 1 and 2 explained 52% and 20%, respectively. The right projection strongly differentiated NR and R sites according to microbial activities. Among the R sites, the PCA shows significant differences between the Cli site and the Aca and Mf sites. The most likely hypothesis is that the Cli site is still in the process of ecological restoration and that the soil microbial communities have not yet recovered an optimal activity after only a few years after the cessation of mining, contrary to the Aca and Mf sites which have a functional diversity indicating a natural state. The second major finding of this study, as shown in the principal component analysis (Figure 2), is the overall lack of recovery of biological activity and functional diversity, indicators of damaged soils, for the unrestored sites Sp1 and Sp2 after more than 18 years of mining cessation. Without the establishment of a forest cover, it seems from these results that the soil cannot regain functional diversity within a short period of time to ensure the proper functioning of biogeochemical cycles. These results are in accordance with those of Li et al. [23], who found that different carbon sources due to vegetation restoration patterns significantly influenced the metabolic activity and functional diversity of the microbial community in the soil [23]. Further, Bisseger et al. [36] showed that plant type and combination play a critical role in the development of the microbial communities present in treatment wetlands, in particular by modifying the nature and quality of organic matter.

As shown in Table 5, the correlations between the different parameters related to microbial activity showed overall positive and significant correlations between carbon, nitrogen, and phosphorus mineralization and functional richness indicators. However, the main point concerned the positive correlations between the microbial biomass (MBC) and the totality of the parameters related to the mineralization of organic matter and the parameters related to catabolic diversity. While MBC had already been shown to be a good indicator of enzymatic activities in soils [40], these results could also confirm the relevance of this indicator for estimating the catabolic diversity of microbial communities in sites undergoing ecological restoration.

### 3.4. Effect of Restoration with Legumes

In French Guiana, the ecological rehabilitation of mining sites and their monitoring was relatively recent compared to the gold panning history. For the past 20 years, the protocols put in place had been chosen mainly based on their capacity to cover rehabilitated sites within a few years, and a multitude of methods have been developed and adapted to different types of soils and substrates, including some pioneering ones mostly described by Loubry [4]. Among the many plant species capable of growing on highly anthropized mining plots, fabaceous species have been widely used in French Guiana [4]. Their ability to fix atmospheric nitrogen through rhizospheric symbiosis made these species very competitive [6,7]. *Acacia mangium* was among the most effective, and its ability to restore the soil’s organic matter stock has been demonstrated in previous studies [8,41]. Nevertheless, most assessments of restoration protocols are not standardized and are difficult to implement in tropical systems.

The results presented in this study highlight the contribution of a monoculture of *Acacia mangium* (Aca), and moreover the mixed association of *Clitoria racemosa* and *Acacia mangium* to the recovery of stocks in major macroelements (i.e., C, N, and P) compared to spontaneous vegetation cover (Sp). On the Sp plots, the return of vegetation was still blocked after more than 15 years of rehabilitation, and the microbial biomass was very low. Given the similar trace element and particle size contents in the Sp and Aca plots, it would appear that CNP content after mining was limiting and prevented the return of natural vegetation cover [28]. This improvement in soil quality through the use of legumes is consistent with the literature where several studies have shown that the fabaceous species used for ecological restoration provide the soil with a stock of organic matter of variable quality. Cattanio et al. [42] showed that *Clitoria racemosa* litter was more rapidly mineralized than *Acacia mangium* litter which therefore promotes a rapid return of microbial activity. Although other studies [43,44] have also shown that the productivity of fabaceous plants under ecological restoration conditions is not always indicative of the quality of organic matter, it appears from our results that the quantity of organic matter could be more important than the quality of this organic matter for the return of soil activities, and therefore of the stimulation of microorganisms responsible for the main biogeochemical cycles.

Our results also showed that the combination of *A. mangium* and *C. racemosa* allowed for a significant improvement in C, N, and P compared to an Acacia monoculture alone. Increasing biodiversity made it possible to encourage the development of new microbial communities involved in biogeochemical cycles of macroelements [45] thus encouraging the turnover of organic matter. The soil texture of the Aca and Mf sites was similar; the differences in chemical quality observed could therefore be exclusively linked to the nature of the vegetation cover and its associated microorganisms. The establishment of fabaceous species, particularly in polyculture, seems to be suitable for the restoration of mining sites [2,28]. This result is also encouraging as it proves that the use of local plant species such as *Clitoria racemosa* in ecological restoration of sites is possible and that non-local species such as *Acacia mangium* could be replaced.

## 4. Materials and Methods

### 4.1. Site Description and Soil Sampling

The work was undertaken in French Guiana, South America. The sampled soil sites for this study were rehabilitated after gold mining extraction during decades. According to recent mining operators, exploitation on each of the sites did not last more than a few years, but the difficulty of accessing past records makes it impossible to determine the precise total duration of exploitation on these plots. The main reason for this is that gold miners have been accustomed to returning to old plots that have already been mined, thanks to improvements in mining processes. After closing the alluvial mine pit, rehabilitation consisted of reconstituting the soil excavated during mine exploitation. All excavated soils were then homogenized and flattened with heavy machinery before restoration began [4]. These soils can be considered as an anthroposol, and they have a different structure and texture than natural tropical forest soils.

Six sites were chosen as a function of the restoration method adopted. These six sites were grouped into two categories: three non-restored (NR) sites and three restored (R) sites by an ecological restoration protocol established by Huttel and Loubry during the ecological restoration experimentation campaigns that were carried out in French Guiana in the 1990s. In the absence of a precise register and history, and given the difficulty of using satellite data, it was not possible to determine the surface area of each site. Nevertheless, each site is located within a mining track, which corresponds to a width of approximately 200–300 m around the major riverbed, for several hundred meters in length. The main characteristics of the 6 sampled sites are described in Table 6 below and Figure 3 illustrates several of these sites.

At each site, soil samples were collected from depths 0–10 cm with an auger in April 2016. The sampling for the 6 sites consisted of sampling 5 plots (*n* = 5) per site, with 3 sub-samples per plot that were pooled at the laboratory. The sampling of 5 plots per site was mainly determined by the topography of the terrain, the accessibility of the plots and the feasibility of sampling. Soil samples were immediately sealed in sterile hermetic polyethylene bags for transportation to the Cayenne IRD laboratory. Samples were then dried at ambient temperature (25 °C) (i.e., approximately 3 weeks). These 90 collected samples were then sieved at 2 mm, homogenized, and hermetically sealed at 4 °C until use.

### 4.2. Initial Soil Sample Characterization

Before using soil samples in microcosms, granulometry, pH (1:2.5 soil-to-water), total major element (i.e., C, N, P) and total iron, aluminum, and mercury content, and microbial biomass carbon (using the substrate-induced respiration SIR method) were determined to characterize the main physicochemical and biological properties; this enabled us to assess the main differences between soils.

### 4.3. Soil Granulometry Determination

Granulometry was determined on granules less than 2 mm, and five classes of particles were distinguished according to the NF X 31-107 standard: clay (<2 µm), fine silts (2 to 20 µm), coarse silt (20 to 50 µm), fine sands (0.050 to 0.200 mm) and coarse sands (0.200 to 2 mm). The results were grouped into 3 parts: clay, total silt, and total sand.

### 4.4. Soil Total Carbon, Nitrogen, and Phosphorous Measurement

Soil total carbon (Ctot) and total nitrogen (Ntot) were determined by the Dumas method (NF ISO 13878) by gas chromatography with a thermal conductivity detector (NA 1500 series 2 CARLO-ERBA, Val-de-Reuil, France).

Total phosphorus (Ptot) was determined after the acid digestion of soil samples, as previously described by the ammonium molybdate and acid ascorbic method [30]. The color produced by phosphomolybdenum (PMB) was measured by colorimetry at 885 nm (spectrophotometer Genesys 10 µv Scanning, Thermo Scientific, Armand Asnières-sur-Seine, France).

Water extractions were performed to measure total dissolved organic carbon concentrations (DOC), with a Shimadzu TOC-500 apparatus (Shimadzu, Kyoto, Japan). One gram of soil was shaken for 24 h in a polypropylene centrifuge tube with 10 mL of ultrapure water. The suspensions were centrifuged at 2000 rpm for 10 min, and supernatants were filtered at 0.2 µm (PTFE, VWR©) and directly analyzed on the TOC analyzer.

### 4.5. Soil Total Iron, Aluminum and Mercury Content

For total Fe and Al analyses by inductively coupled plasma optical emission spectroscopy (ICP-OES, Spectroblue, Elancourt-France), the samples were first ground to 63 µm then an acid digestion was performed in cleaned Teflon digestion tubes (SCP Science, France) using a hot block (Digiprep MS, SCP Science, Villebon sur Yvette-France) with two digestions (HNO_3_ (VWR ref. 83872.330-HF) (Sigma-Aldrich, Saint Louis, MO, USA, ref. 339261-100ML) (3:1 mL) and ultrapure HCL (VWR ref. 83871.290)).

Mercury concentrations in soil samples (ground to 63 µm) were determined directly by thermal decomposition atomic absorption spectrometry after gold amalgamation using an automatic mercury analyzer (AMA 254) with a detection limit of 0.01 ng.

### 4.6. Soil Microbial Biomass

To estimate the catabolic diversity of microbial communities, we used the EcoPlate device (ECOLOG, Hayward, CA, USA). It was first necessary to determine the biomass of microorganisms in the samples and then inoculate the EcoPlate at constant biomass.

The application of the substrate-induced respiration technique (SIR) was applied to determine the total microbial biomass in accordance with Anderson and Domsch [46]. The CO_2_ concentrations were analyzed using micro gas chromatography (micro-GC 490, Agilent, les Ulis, France). The CO_2_ evolution rate (SIR, μLCO_2_-C g^−1^ soil h^−1^) at each sampling time was then calculated after subtracting the background CO_2_ concentration. Using the SIR method, soil microbial biomass C (SIR-biomass C, μgC g^−1^ soil) was calculated using the following equation from Anderson and Domsch [46]:SIR-biomass C () = SIR () × 40.04 + 0.37 (1)

### 4.7. Catabolic Diversity Using Biolog EcoPlates

The community-level physiological profiles were assessed using the Biolog EcoPlateTM system (Biolog Inc., Hayward, CA 94545, USA). According to Garland [47], microbial cell suspensions for the inoculation of the Biolog EcoPlateTM were prepared by agitating 1 g of soil in 10 mL of NaCl (9 g L^−1^) for 30 min at 300 rpm and then centrifuging the suspensions (3000 rpm, 10 min) to eliminate soil particles. The absorbance at 595 nm was directly measured by a plate spectrometer (Dynex Osy MR). The average well color development (AWCD) was calculated, in accordance with Garland and Mills [47]. Potential functional richness (PFR) was estimated with the average number of positive wells (DO > 0.15). The catabolic diversity determining substrate utilization capacity was assessed by dividing the average number of positive substrates by the total number of substrates (31 different substrates). The Shannon Index (*H*) and the regularity index (*EH*) were calculated using the following formulas: (2)$$H=-\sum_{i=1}^{N}Pi\;\left(LnPi\right)$$
(3)$$EH=\frac{H}{Ln\;S}$$
where *Pi* is the ratio of the activity of a particular substrate to the sum of the activities of all substrates and *S* is the richness of the substrate.

### 4.8. Mineralization Rate of Microbial Communities

To quantify carbon, nitrogen and phosphorus mineralization activities in soil samples, fifty milliliters of sterile hermetic glass plasma bottles were filled with 4 g of each soil sample. Soil samples were then brought to 70% of their holding capacity with sterile ultrapure water, and soil humidity was kept constant with sterile ultrapure water. The obtained microcosms were mixed with sterile spatulas, sealed with butyl/PTFE shield caps (VWR©) and incubated at 28 °C for 10 days using the same procedure as described in a previous study [28]. To avoid anaerobiosis conditions, the microcosms were regularly aerated under a laminar flow hood in sterile conditions (PSM Optima 12; ADS Laminaire).

The mineralized total carbon (Cmin) was evaluated by measuring soil CO_2_ produced during the experimental setup. Every three days, the release of CO_2_ was measured by micro gas chromatography (Agilent, Micro GC 490, PPQ column). The value of Cmin was expressed as µg-C CO_2_ g^−1^ of soil day^−1^.

The mineralized nitrogen (Nmin) was determined by measuring the production of mineral N (NH_4_^+^ and NO_3_^−^) during incubation. The NH_4_^+^ and NO_3_^−^ contents were measured at day 0 and day 30 by indophenol (ISO 7150-1) and dimethyl-2.6-phenol methods (DIN 38405-9), respectively. The net ammonification and nitrification rates were calculated as the difference in N-NH_4_^+^ and N-NO_3_^−^ contents before and after incubation (i.e., day 0 and day 30). The mineralization rate of total organic N (Nmin, µg-N g^−1^ of soil day^−1^) was estimated by summing the ammonification and nitrification rates. 

The mineralized phosphorus (Pmin, µg-P g^−1^ of soil day^−1^) was determined by measuring the production of mineral phosphorus during incubation according to Murphy and Riley [48]. The color produced by phosphomolybdenum (PMB) was measured with a spectrophotometer at 885 nm.

### 4.9. Statistical Analysis

To determine the effect of vegetation cover on the microbial activities, parametric tests were performed. The normality of data distributions and equal variance between treatments were tested using the Shapiro test and Bartlett’s test, respectively. To study the effect of the restoration processes, we conducted one-way ANOVA, following by Tukey’s HSD multiple comparison method. Pearson correlation was performed between the parameters of the dataset to highlight the relationships between the variables.

To analyze the effect of the different restoration processes on the global variability of the dataset, the most important results including microbial activities, soil composition and functional diversity of microbial communities were submitted to a principal component analysis (PCA) using the ADE4 package on R. The significance of each explanatory variable was tested using a Monte Carlo permutation test. The R software was used for all statistical analyses (R version 3.3.2 (31 October 2016)).

## 5. Conclusions

This study, carried out on soil samples from rehabilitated Guyanese mining sites, showed an improvement in the soil structure and functionalities of the biogeochemical cycles of C, N and P depending on the density and nature of the vegetation cover. For all microbial indicators measured in this study, ecological restoration under controlled conditions with Fabaceae species showed significantly higher beneficial effects for soil microbial activities than for sites not actively restored. The results also illustrated that Fabaceae associations are more effective than monocultures in restoring soil microbial activities.

This study also revealed a possible modification of the catabolic diversity of microbial communities at rehabilitated sites, which would confirm the hypothesis that ecological restoration directly influences soil microorganism activities. However, the results presented in this study do not make it possible to determine whether the modification of catabolic diversity is linked to an alteration in the genetic structure of communities. Moreover, these results cannot be generalized to all rehabilitated mining sites because of the very wide range of rehabilitation and restoration protocols adopted in French Guiana and the Amazon basin but have shown promising results in assessing the quality of ecological restoration. Indeed, the microbial and biogeochemical approach, based on catabolic diversity of microbial communities, seems to be a solid basis for assessing the quality of ecological rehabilitation protocols for Guiana’s mining sites. Of the indicators presented in this study, microbial biomass carbon (MBC) was found to be the best overall indicator of both biogeochemical cycles and catabolic diversity.

## Figures and Tables

**Figure 1 molecules-27-01768-f001:**
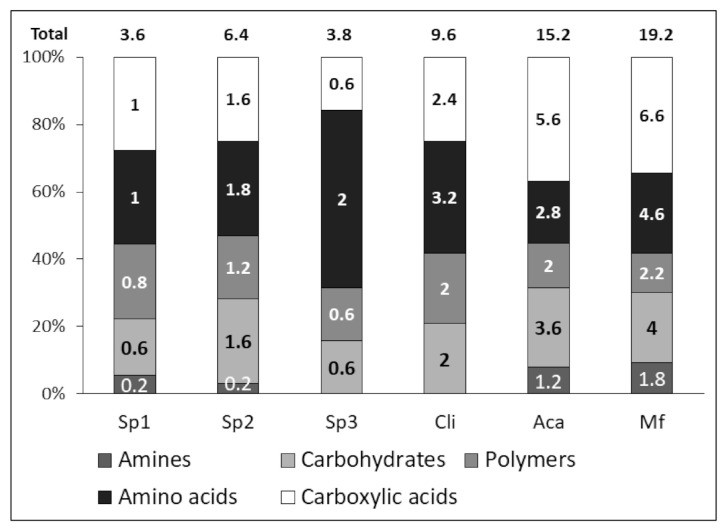
Percentage degradation of substrates according to biochemical categories for the various rehabilitated soils. Sp1, Sp2 and Sp3 are non-restored mining sites. Cli is a restored site with *C. racemosa*, Aca is a restored site with *A. mangium*, and Mf is a mixed fabaceous species. The value in each bar of the histograms is the average value of positive wells (*n* = 5) for the different substrate categories. The value at the top of each histogram corresponds to the average value of positive substrates for each of the sites studied (*n* = 5).

**Figure 2 molecules-27-01768-f002:**
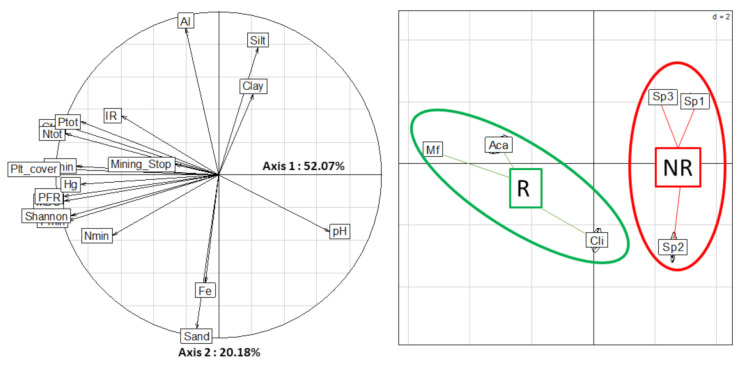
Principal component analysis on the most explanatory variables and representative of the variability of the database. (**Left**): projection of dataset variability plotted on a factorial map of the first two discriminating axes according to site factor: Sp1, 2 and 3. Spontaneous vegetation, Cli: monoculture of *C. racemosa*, Aca: monoculture of *A. mangium*, Mf: mixed fabaceous culture. (**Right**): correlation circles plot with variable vectors (MBC, C/N/P mineralization, total CNP, total Al/Fe/Hg, clay, silt, sand, pH, PFR, IR, Shannon Index, Mining_stop—number of years after mining stopped, Plt_cover—approximate percentage of vegetation cover (according to the classification of Tropical Ecosystem Environment Observations) for each respective factor. Eigen values 52.07%, 20.18% for axes 1 to 2, respectively. Randtest: simulated *p*-value: 0.001.

**Figure 3 molecules-27-01768-f003:**
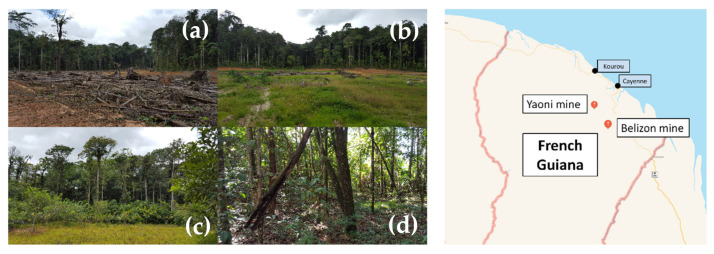
Illustration of some examples of different stages of ecological restoration on former Guyanese gold mining sites and geographical location of the Yaoni and Belizon mines in French Guiana. Non-restored site Sp1 (**a**), non-restored site Sp2 with herbaceous species after 3 years (**b**), ecological restoration with fabaceous species after 3 years ((**c**), Cli), ecological restoration with fabaceous species after 18 years ((**d**), Mf). The pictures were taken during a second sampling campaign in April 2016 at sites undergoing ecological restoration.

**Table 1 molecules-27-01768-t001:** Ctot, Ntot, Ptot: total C, N and P; DOC: total organic carbon; MBC: microbial biomass carbon; total Fe, Al, Hg: total iron, aluminum and mercury content. NR: non-restored sites, R: restored sites. Sp1, Sp2 and Sp3 are non-restored mining sites. Cli is a restored site with *C. racemosa*, Aca is a restored site with *A. mangium,* and Mf is a mixed fabaceous species (*n* = 5, mean ± SD). For each parameter, values followed by different letters differ significantly with *p* < 0.05 with Tukey’s HSD test.

Sites	Restored	Total											
		C_tot_	DOC	N_tot_	P_tot_	C/N	pH-H_2_O	Fe	Al	Hg	Clay	Silt	Sand
		g·kg^−1^	g·kg^−1^	g·kg^−1^	g·kg^−1^			g·kg^−1^	g·kg^−1^	µg·kg^−1^	%	%	%
Sp1	NR	6.01b ± 0.32	3.88ab ± 0.04	0.40b ± 0.03	0.21ab ± 0.04	15.02b ± 0.30	5.26ab ± 0.23	65.15b ± 3.17	53.88b ± 1.53	0.23a ± 0.01	46.4c ± 3.6	41.5b ± 2.8	3.3a ± 0.2
Sp2	NR	3.52a ± 0.37	2.47a ± 0.23	0.14a ± 0.02	0.13a ± 0.01	25.14c ± 1.08	5.3b ± 0.13	66.2b ± 2.1	16.4a ± 0.6	0.27a ± 0.02	7.5a ± 2.7	21a ± 2	71.1d ± 1.6
Sp3	NR	9.55b ± 0.73	5.84b ± 0.35	0.45b ± 0.01	0.27b ± 0.07	21.2c ± 1.27	4.66ab ± 0.07	44.56a ± 0.77	41.33b ± 1.77	0.23a ± 0.07	22.2b ± 5.1	65.8c ± 3.2	11.8b ± 1.2
Cli	R	7.01b ± 0.83	5.0b ± 0.4	0.40b ± 0.04	0.17a ± 0.02	17.52b ± 1.35	5.0ab ± 0.03	113.3c ± 4.2	25.5a ± 2.15	0.28a ± 0.03	21.3b ± 1.6	24.8a ± 3.7	53.9c ± 2.5
Aca	R	14.11c ± 1.2	10.55c ± 0.62	1.12c ± 0.07	0.37b ± 0.12	12.6a ± 0.54	4.21a ± 0.07	53.69ab ± 4.15	45.31b ± 6.51	0.48b ± 0.04	18.4b ± 2.5	35.6b ± 3.9	41.6c ± 4.4
Mf	R	22.88d ± 1.5	15.58d ± 1.41	1.81d ± 0.10	0.76c ± 0.15	12.64a ± 0.41	4.66ab ± 0.11	71.02b ± 1.37	43.76b ± 1.21	0.41b ± 0.05	19.8b ± 3.8	32.4b ± 4.9	36.4c ± 5.9

**Table 2 molecules-27-01768-t002:** MBC: microbial biomass carbon; Cmin (SIR); Cmin, NH_4_^+^-Nmin, NO_3_^−^-Nmin, Pmin: organic C mineralization rate, mineralized NH_4_^+^-N, mineralized NO_3_^−^-N, total N mineralization rate, organic P mineralization rate. All mineralization rates are expressed in micrograms per gram of soil per day. NR: non-restored sites, R: restored sites. Sp1, Sp2 and Sp3 are non-restored mining sites. Cli is a restored site with *C. racemosa*, Aca is a restored site with *A. mangium*, and Mf is a mixed fabaceous species (*n* = 5, mean ± SD). For each parameter, values followed by different letters differ significantly with *p* < 0.05 with Tukey’s HSD test.

Sites	Restored	MBC	C_min_	NO_3_^−^-N_min_	NH_4_^+^-N_min_	Ntot_min_	P_min_
		mg kg^−1^	µg g^−1^day^−1^	µg g^−1^day^−1^	µg g^−1^day^−1^	µg g^−1^day^−1^	µg g^−1^day^−1^
Sp1	NR	224.5a ± 13.2	4.45a ± 0.32	0.008a ± 0.001	0.13a ± 0.06	0.14a ± 0.07	0.017a ± 0.003
Sp2	NR	390.5a ± 77.3	4.96a ± 1.1	0.07b ± 0.006	0.8b ± 0.2	0.88c ± 0.2	0.024a ± 0.002
Sp3	NR	312.6a ± 10.1	5.77a ± 0.37	0.004a ± 0.0003	0.45ab ± 0.05	0.46b ± 0.05	0.03b ± 0.002
Cli	R	845.2b ± 38.5	35.37b ± 1.4	1.56d ± 0.35	1.15c ± 0.15	2.7e ± 0.44	0.047c ± 0.007
Aca	R	909b ± 69	39.14bc ± 1.9	0.33c ± 0.12	1.2c ± 0.32	1.55d ± 0.33	0.058cd ± 0.006
Mf	R	1103b ± 88	46.58c ± 1.5	0.35c ± 0.11	1.7d ± 0.14	2.03e ± 0.07	0.065d ± 0.008

**Table 3 molecules-27-01768-t003:** AWCD: average well color development, PFR: potential functional richness, catabolic diversity, H: Shannon Index, EH: regularity index, NR: non-restored sites, R: restored sites. Sp1, Sp2 and Sp3 are non-restored mining sites. Cli is a restored site with *C. racemosa*, Aca is a restored site with *A. mangium*, and Mf is a mixed fabaceous species (*n* = 5, mean ± SD). For each parameter, values followed by different letters differ significantly with *p* < 0.05 with Tukey’s HSD test.

Sites	Restored	AWCD	PFR	Catabolic Diversity	H	E_H_
Sp1	NR	0.08a ± 0.01	3.6a ± 0.9	0.12a ± 0.02	1.2a ± 0.11	0.76a ± 0.05
Sp2	NR	0.18b ± 0.1	6.6ab ± 2.1	0.21b ± 0.07	1.9ab ± 0.47	0.53a ± 0.25
Sp3	NR	0.09a ± 0.02	3.8a ± 0.8	0.12a ± 0.03	1.3a ± 0.44	0.88b ± 0.036
Cli	R	0.34c ± 0.13	9.6b ± 1.9	0.31c± 0.06	2.1b ± 0.12	0.92b ± 0.015
Aca	R	0.32c ± 0.1	15.2c ± 3.2	0.49d ± 0.11	2.6c ± 0.22	0.95b ± 0.027
Mf	R	0.49d ± 0.09	19.2c ± 1.1	0.62d ± 0.04	3.3c ± 0.11	0.98b ± 0.01

**Table 4 molecules-27-01768-t004:** Effects of vegetation and level of restoration in a one-way ANOVA. Bold values are significant (*p* < 0.05), “*” indicates a *p*-value 0.01 < *p* < 0.05, “**” indicates a *p*-value 0.001 < *p* < 0.1.

		Sites	Level of Restoration
	Df	5	1
Total organic carbon	F-value	**4472 ****	**25 ****
Total nitrogen		**1529 ****	**22 ****
Total phosphorus		**224 ****	**11 ***
Microbial biomass carbon		**1330 ****	**322 ****
Carbon mineralization		**480 ****	**20 ****
Nitrate mineralization (NO^3−^-N_min_)		**67 ****	**18 ****
Ammonium mineralization		**49 ****	**60 ****
Phosphorus mineralization		**58 ****	**94 ****
Average well color development (AWCD)		**15 ****	**47 ***
Potential functional richness		**56 ****	**60 ****
Catabolic diversity		**5 ****	**60 ****
Shannon Index		**37 ****	**44 ****
Regularity index		**11 ***	**17 ***

**Table 5 molecules-27-01768-t005:** Pearson correlation coefficients (*n* = 30) between microbial biomass carbon (MBC), mineralization rates and microbial catabolic diversity parameters; bold values are significant at *p* < 0.05.

Variables	MBC	C_min	NO_3__min	NH_4__min	N_tot__min	P_min	PFR	Diversity	H	E_H_	AWCD
MBC	**1.00**	**0.80**	**0.50**	**0.92**	**0.84**	**0.93**	**0.89**	**0.89**	**0.85**	**0.64**	**0.83**
C_min	-	**1.00**	0.11	**0.79**	**0.53**	**0.75**	**0.82**	**0.82**	**0.82**	**0.47**	**0.74**
NO_3__min	-	-	**1.00**	**0.40**	**0.84**	0.36	0.20	0.20	0.22	0.35	0.35
NH_4__min	-	-	-	**1.00**	**0.83**	**0.80**	**0.77**	**0.77**	**0.77**	**0.59**	**0.72**
N_tot__min	-	-	-	-	**1.00**	**0.69**	**0.57**	**0.57**	**0.58**	**0.56**	**0.64**
P_min	-	-	-	-	-	**1.00**	**0.93**	**0.93**	**0.89**	**0.48**	**0.85**
PFR	-	-	-	-	-	-	**1.00**	**1.00**	**0.95**	**0.43**	**0.90**
Diversity	-	-	-	-	-	-	-	**1.00**	**0.95**	**0.43**	**0.90**
H	-	-	-	-	-	-	-	-	**1.00**	0.28	**0.87**
E_H_	-	-	-	-	-	-	-	-	-	**1.00**	0.35
AWCD	-	-	-	-	-	-	-	-	-	-	**1.00**

**Table 6 molecules-27-01768-t006:** Main characteristics of the 6 sampled sites.

Sites	Sp1	Sp2	Sp3	Cli	Aca	Mf
level of restoration	Non-restored	Non-restored	Non-restored	Restored	Restored	Restored
Pioneer species	Herbaceous species	*Cyperus* spp. and *Carex* spp.	*Lycopodiella* spp.	*Clitoria racemosa*	*Acacia mangium*	*C. racemosa* and *A. mangium*
Type of vegetation cover	Almost bare soil	Grassland	Grassland	Coppiced forest	Dense forest	Dense forest
End of mining (year)	1998	1998	2013	2013	1998	1998
Geographical coordinates	N 04°30.311′/ W 052°26.919′	N 04°30.311′/ W 052°26.919′	N 04°22.509′/ W 052°20.739′	N 04°29.860′/ W 052°26.966′	N 04°29.860′/ W 052°26.966′	N 04°29.860′/ W 052°26.966′

## Data Availability

Data are available by contacting the corresponding authors.

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
