# Peer review of "Positive Effect of Ecological Restoration with Fabaceous Species on Microbial Activities of Former Guyanese Mining Sites"

_molecules, 2022, doi:10.3390/molecules27061768_

Round 1

Reviewer 1 Report

Introduction: Add the significance of soil microorganisms in the carbon, nitrogen and phosphorus cycle. Is there any literature available on restoration, especially with legumes? Need further introduction. In addition, what is the innovation of this article? The authors need highlight it. L63-66: This article does not address this purpose. I suggest to delete it. L64: delete "also" Results: Tables 1-3: Add table headings. Mark the meaning of the abbreviation of the six sites in the table notes. Table 1: There are problems with the layout of the table. The horizontal line should be in the unit line. Please give the results of ANOVA. The determination of Fe and Al is not included in Materials and Methods, and you need to tel usu why are they tested. L144: Please add the results of correlation analysis here. Some results (i.e. Tables 4 and 5) should not be included in the discussion. Show them here. Meanwhile, correlation analysis should be mentioned in the statistical analysis part. Discussion: L150-155: Delete this paragraph. The discussion should focus on legumes, so as to match the topic. The discussion of each soil indicator is too detailed. More needs to be written about the role and changes of microorganisms in the elemental cycles, and the reasons for this, and how they relate to the physical and chemical properties of the soil. Also, why trace elements? It's in the introduction simply, but it's not in the discussion... L162: anthroposols L217-218, Tables 4 and 5: as mentioned above, should be written in the results. Materials and methods: L325: Experimental design? L337-349: Describe the sites in a table. Fig.3: You may add the abbreviations of the sites in the figure. L382-386: This method is not TOC, but water-soluble carbon, i.e. DOC L404: microbial L416: Same as the heading of 4.7? L424-436: Please add references. L449: Please add correlation analysis. Others: There are many problems about superscript and subscript, mainly in unit, CO2, NH4+-N, NO3--N and so on. Such as in L90-91, 104-105, L108, L180, L396-400, L406, L424-436... The references are badly formatted, they should all be numerical. Such as in L212, L233-234, L257, L262, L299, L309...

Author Response

Dear reviewer,
Thank you for taking the time to review my article and comment on this work. 
Regarding your comments I have made many changes, additions and corrections to the text that meet your expectations. Here is a point-by-point list of the responses to each of your comments. 

Introduction: Add the significance of soil microorganisms in the carbon, nitrogen and phosphorus cycle. Is there any literature available on restoration, especially with legumes? Need further introduction. In addition, what is the innovation of this article? The authors need highlight it.

I have added various paragraphs in the introduction to provide additional information.

L63-66: This article does not address this purpose. I suggest to delete it.

I agree, this goal was perhaps beyond the interpretation of the results of this article.

L64: delete "also" . Done.

Results: Tables 1-3: Add table headings. Mark the meaning of the abbreviation of the six sites in the table notes.

I have added the descriptions of the abbreviations of the study sites in the table legend.

Table 1: There are problems with the layout of the table. The horizontal line should be in the unit line. I have reformatted the table

Please give the results of ANOVA. It was an oversight on my part, I added the test results.

The determination of Fe and Al is not included in Materials and Methods, and you need to tel usu why are they tested. It was an oversight on my part, I added the determination in the materials and methods.

L144: Please add the results of correlation analysis here. Some results (i.e. Tables 4 and 5) should not be included in the discussion. Show them here. I have changed the placement of tables 4 and 5 to the results section and added paragraphs to describe the different tables.

 Meanwhile, correlation analysis should be mentioned in the statistical analysis part. I added the correlation analysis in statistical part.

Discussion: L150-155: Delete this paragraph. done

 The discussion should focus on legumes, so as to match the topic. The discussion of each soil indicator is too detailed. More needs to be written about the role and changes of microorganisms in the elemental cycles, and the reasons for this, and how they relate to the physical and chemical properties of the soil. I believe that the description of each of the indicators is important for the overall understanding of the theme and the issues. In addition, the article is more focused on highlighting microbial activities in relation to ecological restoration modalities and I do not seek to develop the relationships with trace elements since the latter are not in high concentrations in soils.

I have taken your point into consideration and added a whole paragraph in the discussion regarding the use of legumes, which are central to the topic and were indeed missing from the discussion.

Also, why trace elements? It's in the introduction simply, but it's not in the discussion...

As the ecological restoration experiments were carried out on mining sites, my collaborators and I wanted to measure trace metals to see if there were any links with microbial activities. However, the levels of pollutants are generally only higher than in natural tropical soils.

L162: anthroposols . done

L217-218, Tables 4 and 5: as mentioned above, should be written in the results. I made the changes

Materials and methods: L325: Experimental design? I prefer to keep the original format of material and method

L337-349: Describe the sites in a table. This is indeed a very good idea. I have added a table summarising the main properties of the sampled soils.

Fig.3: You may add the abbreviations of the sites in the figure. I added the abbreviations

L382-386: This method is not TOC, but water-soluble carbon, i.e. DOC I corrected this mistake.

L404: microbial L416: Same as the heading of 4.7? I corrected this mistake.

L424-436: Please add references. I added.

L449: Please add correlation analysis. I added the correlation analysis.

Others: There are many problems about superscript and subscript, mainly in unit, CO2, NH4+-N, NO3--N and so on. Such as in L90-91, 104-105, L108, L180, L396-400, L406, L424-436...  I corrected those mistakes.

The references are badly formatted, they should all be numerical. Such as in L212, L233-234, L257, L262, L299, L309... I corrected those mistakes.

Best regards,

Reviewer 2 Report

This manuscript examines the abiotic and biotic soil properties of previously worked alluvial gold deposits with and without restoration efforts, describing differences in microbial biomass, metabolic activity with respect to bulk nutrients (C, N, P) and more nuanced catabolic diversity of a wide range of possible substrates. This is a well written paper on a subject that important to French Guiana but also is global application and interest as former mine site restoration and soil restoration in general as mineral resource extraction and resulting degraded soils has grown exponentially over the last century.

Overall, I quite like this paper, and mostly I have minor edit suggestions. Please see comments posted to the manuscript pdf.

I submit this paper can be published after minor revisions.

More significant is:

Figure 1 is confusing at first as all the values add up to a 100% and why that is so, is not immediately clear from the methods/description of the math. There are further comments made to the pdf expanding on this point.

From your method description TOC seems to be more accurately DOC. Please change or defend that usage.

I would have liked to have seen a more encompassing multivariate analysis of your whole data set or mostly whole data set after removing some redundant measures (like PFR and Diversity, one not both) in the form of a PCA or CCA or similar ordination method as you posses a reasonable range of physical measurements (soil pH, nutrients, grain size) and a range of biotic indicators (nutrient mineralization rates, metabolic diversity and MBC) and possibly other variables of interest (years since mining related activity at site, or years in recovery, % vegetation cover, % woody or forest vegetation cover). 

The analysis and comparisons are mostly grouped by method and lack a whole data approach. Table 4 helps leans towards a larger analysis and the factor-by-factor differences do matter but a big picture view from ordination of the whole data could prove insightful and certainly of interest.

Lastly, not a critique per say, but more a suggestion that soil invertebrate biomass and diversity study of the same sites might be of interest and value as well.

Author Response

Dear reviewer,
Thank you for taking the time to review my article and comment on this work. 
Regarding your remarks about minor corrections, I have made many changes, additions and corrections to the text that meet your expectations.
Concerning your major remarks, I have made various modifications.
First of all, after a thorough re-reading and following your remarks, I totally agree with the fact that the BCA analysis of the data related to catabolic diversity should be removed. There are several reasons why I decided to remove this figure. The first is that it did not provide any additional information compared to figure 1 summarising substrate degradation, and compared to the table summarising all indicators of functional richness and diversity. This redundancy made the article more cumbersome without improving the overall message. In addition, the presence of null values in the dataset made it difficult to interpret the results from this perspective. Finally, the Biolog plates allow to visualize the degradation of 31 different substrates (1 plate = 96 wells = 32 wells including one blank * 3 replicates), nevertheless, the objective of this article was to use the indicators of functional richness resulting from the degradation of the substrates and not to focus on the individual degradation of each substrate, which would perhaps require another study and requires a very strong knowledge in biochemistry. For these reasons I preferred to withdraw the analysis. 

I would like to thank you for suggesting that I carry out a global analysis in the form of a PCA. I followed your recommendation and performed a PCA by integrating the microbial activity variables and the soil composition variables in a global way, adding the vegetation cover. I think this figure provides some very interesting information that deepens the article. 

Concerning your last remark about animal diversity measurements, this is obviously a very interesting opening and will certainly be carried out in the near future by local research companies. 

Best regards, 

Round 2

Reviewer 1 Report

This article has been greatly improved after modification. However, there are still some issues that need to be solved:

Key words:

L23: It is recommended to delete “French Guiana”, and add “microbial community” and “catabolic diversity”

Introduction:

L44: Acacia Leucana

Results:

L27: In table 2, font inconsistency, and wrong usage of super/subscripts.

Table4: “Total dissolved organic carbon”instead of “Total organic carbon”. The significance level represented by * should also be added in the table notes. Please keep same decimal places.

Discussion:

L325-326: Duplicate with previous, should be deleted.

L372-375: Table 5 is the same as previous and should be deleted.

Section 3.4: It has supplementary contents that fit the topic well. But you also need to sort out the logic and try to write in several paragraphs.

L315, 392, 421, etc. : The citation format is incorrect. The year of publication should be deleted.

Materials and Methods:

L506: Please add the area of each plot.

L521: Please delete the parentheses after mercury.

L548:3 in HNO3, 3 should be subscripted.

L575: What is 31? Reference number or the total number of substrates?

L564: The citation format is incorrect. The year of publication should be deleted.

L558-563: 2 should be subscripted in CO2. Please check similar mistakes carefully, such as in L593-596 ...

L564: Please delete the parentheses.

Conclusion:

Generally, the conclusion is only one paragraph, and references are not required. Please reorganize and place in the discussion section what is not appropriate for the conclusion.

Author Response

Dear proofreader, 

Please find attached the new version of the article with your comments taken into account. 
Concerning the minor corrections, I confirm that I have taken all your remarks into consideration and that I have done my best to correct all the typos and errors that you have pointed out. 
Concerning table 5, I have nevertheless chosen to keep it and I have moved it to the result part which includes the PCA allowing to describe the global data. I consider that this table is complementary to the BCP and its presence was also one of the major requests of the second reviewer. 

I have clarified part 3.4 of the discussion by structuring my thinking more and adding some information. 
I have added a note on the surface areas of the study sites. 

Best regards